Rice koji reduced body weight gain, fat accumulation, and blood glucose level in high-fat diet-induced obese mice

Yoshizaki Yumiko 1
Kawasaki Chihiro 1
Cheng Kai-Chun 2
Ushikai Miharu 2
Amitani Haruka 2
Asakawa Akihiro 2
Okutsu Kayu 1
Sameshima Yoshihiro 1
Takamine Kazunori 1
Inui Akio 2 inui@m.kufm.kagoshima-u.ac.jp
1 Division of Shochu Fermentation Technology, Education and Research Center for Fermentation Studies, Faculty of Agriculture, Kagoshima University , Kagoshima City , Japan
2 Department of Social and Behavioral Medicine, Kagoshima University Graduate School of Medical and Dental Sciences , Kagoshima City , Japan
Marunaka Yoshinori
Electronic publication date: 2014 Aug 26
Publication date: 2014
Volume: 2
Electronic Location ID: e540
Received 2014 Apr 3; Accepted 2014 Aug 5
Copyright: © 2014 Yoshizaki et al.
Copyright year: 2014
Copyright holder: Yoshizaki et al.
License: This is an open access article distributed under the terms of the Creative Commons Attribution License, which permits unrestricted use, distribution, and reproduction in any medium, provided the original author and source are credited.
License URL: https://creativecommons.org/licenses/by/3.0/

Keywords: GLUT4, High-fat diet, Antiobesity, Mice, Rice koji

Funding: Ministry of Education, Culture, Sports, Science and Technology (MEXT), Japan This study was supported by a Grant-in-Aid for special education research by the Ministry of Education, Culture, Sports, Science and Technology (MEXT), Japan. The funders had no role in study design, data collection and analysis, decision to publish, or preparation of the manuscript.

==============================
Rice koji is considered a readily accessible functional food that may have health-promoting effects. We investigated whether white, yellow, and red koji have the anti-obesity effect in C57BL/6J mice fed a high-fat diet (HFD), which is a model for obesity. Mice were fed HFD containing 10% (w/w) of rice koji powder or steamed rice for 4 weeks. Weight gain, epididymal white adipose tissue, and total adipose tissue weight were significantly lower in all rice koji groups than in the HFD-rice group after 4 weeks. Feed efficiency was significantly reduced in the yellow koji group. Blood glucose levels were significantly lower in the white and red koji groups with HOMA-R and leptin levels being reduced in the white koji group. White and red koji increased glucose uptake and GLUT4 protein expression in L6 myotube cells. These results showed that all rice koji have the anti-obesity or anti-diabetes effects although the mechanisms may differ depending on the type of rice koji consumed.

Introduction

Obesity, which is characterized by an increase in fat mass and body weight, results from a combination of excessive food intake and a lack of physical activity (Carek & Dickerson, 1999). Obesity is a problem in many developed countries because it is often associated with many additional health problems: cancer, hypertension, neurological disorders, and cardiovascular diseases (Spiegelman & Flier, 2001; Formiguera & Cantón, 2004; Steinbaum, 2004). Type 2 diabetes is also related to obesity. The prevalence and incidence of type 2 diabetes is dramatically increasing worldwide in both developed and developing countries. This disease results from the interaction of genetic predispositions such as defective β-cell function and environmental factors such as insulin resistance, which is influenced by lifestyle and the degree of physical activity. Because obesity increases the risk of illness and premature mortality (Barlow et al., 1995; Hu et al., 2001), the prevention or abatement of obesity is important.

Dietary modifications, exercise, behavioral treatment, and drug therapy have been shown to be helpful for treating obesity (Dulloo et al., 1999). Currently, interest in safer diet treatments is increasing, and the trend of consumer preferences for natural foods has been observed (Leonard et al., 2002). Japan has many traditional fermented foods such as sake, miso, soy sauce, and shochu. These products have attracted worldwide attention as foods that might promote longevity because of their superior antioxidant properties and functional nutrients compared to non-fermented ingredients (Murooka & Yamashita, 2008).

Koji, which is a solid-state culture of koji molds on cereal grains such as rice and barley, is the commonly used material in these products. Because koji mold produces many types of enzymes and secretes them as it grows, koji is important for the digestion of plant materials like starch, protein, and lipid during the manufacture of fermented foods. These enzymes not only are useful for fermentation but also contribute to the enhancement of the functional nutrient content of fermented products (Koseki et al., 1998). Furthermore, it has also been confirmed that koji mold produce the secondary metabolite as functional compounds such as kojic acid and pyranonigrin-A (Miyake et al., 2008; Choi et al., 2012).

Koji is used often to prepare fermented foods in eastern-Asian countries. Yellow koji mold (Aspergillus oryzae) and white koji mold (Aspergillus kawachii) have been used in Japan. Yellow koji mold has been used in sake, soy source, and miso, whereas white koji mold were used exclusively in the manufacture of shochu. Red koji mold (Monascus spp.) has been used as the source of fermented foods in China, Japan, Taiwan, and Indonesia. Red koji mold is well known as the fungus that produces a variety of functional chemical compounds such as monacolin K and γ-aminobutyric acid (GABA) (Su et al., 2003). Although fermented foods have been well studied, the different effects of various types of koji have not been reported.

We hypothesize that rice koji has antiobesity effects because the compounds from koji are presumably responsible for part of the functions of fermented foods (Yamada et al., 1998). Thus, the objective of this study was to evaluate the antiobesity activity of various types of rice koji in an animal model with high-fat diet (HFD)-induced obesity and to elucidate the effect of glucose homeostasis.

Materials and Methods

Animals and cell line

CE-2 as a normal diet and HFD composed of 60% of calories from lipids were purchased from CLEA Japan, Inc. (Tokyo, Japan) and Research Diets Inc. (New Brunswick, NJ, USA), respectively. Male C57BL/6 mice were purchased from CLEA Japan, Inc. at 7 weeks of age. After arrival, each mouse was housed in an individual cage and maintained on a 12-h/12-h light-dark cycle in a temperature-controlled room (22 ± 2 °C, 50% ± 10% humidity). The mice were allowed free access to water and a normal diet for acclimatization (1 week). After acclimatization, the mice were divided into 5 body weight-matched groups (n = 12 per group) as follows. Group-I (CE-rice) was fed CE-2; Group-II (HFD-rice), an HFD containing 10% (w/w) steamed rice powder; Group-III (HFD-Y), an HFD containing 10% (w/w) yellow koji powder; Group-IV (HFD-W), an HFD containing 10% (w/w) white koji powder; and Group-V (HFD-R), an HFD containing 10% (w/w) red koji powder. Body weight and food intake were measured every week. This study was approved by the ethical committee for animal experimentation at Kagoshima University Graduate School of Medical and Dental Sciences (Permit Number: MD10122). The L6 rat skeletal muscle cell line was obtained from American Type Culture Collection Inc. (ATCC, Rockville, MD, USA).

Preparation of rice koji

All rice koji were prepared in our laboratory. The seed cultures of Aspergillus oryzae and Aspergillus kawachii were purchased from Kawauchi Genichiro shoten (Kagoshima, Japan). The seed culture of Monascus anka was purchased from Akita Konno Co., Ltd. (Akita, Japan). Solid-state fermentation was performed to prepare koji as described in our previous paper (Yoshizaki et al., 2010a; Yoshizaki et al., 2010b). The koji preparation was freeze-dried, ground to powder, and stored at −80 °C until use.

Analysis of nutrient components

Moisture, protein, fat, ash, carbohydrate, and energy were analyzed by Japan Food Research Laboratories (Tokyo, Japan). Moisture and protein were measured by heating-drying method and Kjeldahl method, respectively. Fat and ash were measured gravimetrically after the extraction and the incineration, respectively. Carbohydrate was calculated by subtracting the amount of moisture, protein, fat, and ash from 100. The energy values were calculated by means of Atwater’s factors (protein and carbohydrate 4 kcal/g and fat 9 kcal/g, respectively). Glucose was measured according to the method described in our previous study (Okutsu et al., 2012). High-performance liquid chromatography (HPLC) consisted of a COSMOSIL Sugar-D column (i.d., 4.6 × 250 mm; Nacalai Tesque, Inc., Kyoto, Japan) and a refractive index detector (Shimadzu Co., Kyoto, Japan). The mobile phase consisted of CH3CN:H2O (75:25) at a flow rate of 1.0 ml/min. The column temperature was maintained at 40 °C. Citric acid was analyzed using HPLC consisting of an ion-exclusion column (Shim-pack SCR-102H; i.d., 8 × 300 mm × 2; Shimadzu Co.) and post-column pH-buffered electroconductivity detection (Shimadzu Co.). The mobile phase consisted of 4 mM of p-toluenesulfonic acid solution at a flow rate of 0.8 ml/min. A reaction mixture of 4 mM p-toluenesulfonic acid with 16 mM bis-tris and 80 µM ethylenediaminetetraacetic acid disodium salt (EDTA) was used as post-column reagent at a flow rate of 0.8 ml/min. The column temperature was maintained at 50 °C.

Blood sample analysis and isolation of tissue material

After 4 weeks of feeding on the respective diets, mice were fasted for 4 h. Then, blood samples were obtained from the orbital sinus under diethyl ether anesthesia at the end of the experiment. Mice were euthanized by cervical dislocation. Immediately after, liver, visceral white adipose tissue, epididymal white adipose tissue, and brown adipose tissue were removed and weighed. The entire sampling procedure was done in less than 2 min. Blood samples were collected and immediately centrifuged at 1,500 × g for 10 min at 4 °C. The serum was sampled and stored at −80 °C until analysis. Concentrations of total cholesterol (TC), high-density lipoprotein (HDL) cholesterol, triglyceride (TG), and glucose were measured using a clinical chemistry reagent kit (Cholesterol E test, HDL-Cholesterol test, and Triglyceride G test; Wako Pure Chemical Industries, Ltd., Osaka, Japan). Blood glucose was measured with the Nipro Freestyle Freedom (Nipro Co., Osaka, Japan). Concentrations of insulin and leptin were measured using an enzyme immunoassay (Mouse Insulin kit and Mouse leptin kit; Morinaga Institute of Biological Science, Inc., Yokohama, Japan). The adiponectin concentration was evaluated by a sandwich ELISA system (Mouse/Rat adiponectin ELISA kit; Otsuka Pharmaceutical Co. Ltd., Tokyo, Japan). Homeostasis model assessment of insulin resistance (HOMA-R) was calculated as (fasting glucose level (mg/dl) × fasting insulin level (ng/ml))/450.

Total RNA isolation and real-time PCR

Total RNA in the liver was prepared from frozen tissue using the NucleoSpin RNA II kit (Macherey-Nagel Ltd., Düren, Germany). The cDNA was synthesized from 1 µg of total RNA with the Prime Script RT reagent kit (Takara Bio Inc., Tokyo, Japan). Real-time PCR was performed using SYBR Premix ExTaq (Takara Bio Inc.) on a Thermal Cycler Dice real-time system (Takara Bio Inc.) for 40 cycles of 95 °C for 30 s and 60 °C for 30 s. Commercially available, prevalidated primer pairs were purchased from the Perfect Real-time primer Support System (Takara Bio Inc.). β-Actin (GenBank accession No. NM_007393.3) was used as endogenous control for all target genes. Fatty acid synthase (FAS; GenBank accession No. NM_007988.3), acetyl-coenzyme A carboxylase-α (ACACα; GenBank accession No. NM_NM_133360.2), acetyl-coenzyme A carboxylase-β (ACACβ; GenBank accession No. NM_133904.2), carnitine palmitoyltransferase 2 (Cpt2; GenBank accession No. NM_009949.2), acyl-Coenzyme A dehydrogenase, very long chain (Acadvl; GenBank accession No. NM_017366.3), acyl-Coenzyme A dehydrogenase, long-chain (Acadl; GenBank accession No. NM_007381.4), acyl-Coenzyme A dehydrogenase, medium chain (Acadm; GenBank accession No. NM_007382.5), and peroxisome proliferator-activated receptor-α (PPARα; GenBank accession No. NM_001113418.1) were the target genes. Gene expression results are expressed as the expression ratio relative to β-actin gene expression, which were determined according to the manufacturer’s instructions. Product specificity was verified by melting curve analysis.

Glucose uptake assay in L6 myotube cells

Steamed rice, white koji, or red koji powder (5 g) was added to 25 ml deionized water and homogenized at 11,000 rpm for 1 min. The homogenate was centrifuged at 21,500 × g for 10 min, and the supernatant was used as the extract. The extract was diluted in deionized water for use in the assay. The dilution ratios of these extracts are shown in the figures as the dilution factor (DF). Insulin (0.1 µM) was used in place of the extract as a positive control.

For the glucose uptake assay, monolayers of L6 myotube cells were grown in DMEM/high glucose medium containing 4.5 g/L glucose (HyClone; Thermo Fisher Scientific Inc., Logan, UT) in a humidified 5% CO2 incubator at 37 °C. Cells (1 × 105/ml) were seeded in 10-cm petri dishes and grown to 90% confluency. The cells were washed 3 times in phosphate-buffered saline (PBS). The culture medium was replaced for 3 h with DMEM without glucose medium (Invitrogen Co., Carlsbad, CA, USA), and the cells were trypsinized. A centrifugation step was performed to remove trypsin, and the cells were resuspended in DMEM without glucose medium at 2 × 106 cells/ml. This culture medium (0.5 ml containing 1 × 106 cells) was transferred to each test tube, and 0.5 ml DMEM without glucose medium and the extract or insulin solution (0.1 ml) were added to the tube. After preincubation at 37 °C for 5 min, 10 µl of 20 mM 2-(N-[7-nitrobenz-2-oxa-1,3-diazol-4-yl]amino)-2-deoxyglucose (2-NBDG; Invitrogen Co.) was added (Zou, Wang & Shen, 2005). Test tubes were incubated at 37 °C with 5% CO2 for 30 min. The 2-NBDG uptake reaction was stopped by removing the incubation medium and washing the cells twice with pre-chilled PBS. Each test tube was subsequently resuspended in 500 µl pre-chilled PBS and measured on a fluorescence spectrometer (Hitachi F-2500; Hitachi, Ltd., Tokyo, Japan) with excitation and emission set at 480 and 520 nm, respectively. All data presented were obtained from at least 3 separate cell preparations.

Western blotting analysis

The koji extract was the same sample as in the glucose uptake assay. Monacolin K, manscin, or citrate-Na buffer (pH 7.4) were also prepared to exposure for cells; monacolin K solutions (1–1000 µM), monascin solution (5–20 µM), and citrate-Na buffer (pH 7.4) (3–150 mM), respectively. Cells (1 × 105 cells) were grown in 10-cm petri dishes with 5 ml DMEM/high glucose medium and 0.1 ml samples for 14 h. The L6 myotube cells were washed 3 times in PBS, trypsinized, and the trypsin was removed by centrifugation. Protein extracts were prepared by homogenizing the tissue in 50 mM Tris–HCl (pH 7.6) and 0.1% Triton X-100 supplemented with a protease inhibitor cocktail (F. Hoffmann-La Roche Ltd., Mannheim, Germany) and centrifuged at 21,500 × g for 30 min. The supernatant was used as the crude extract. The protein concentrations of extracts were determined using the Coomassie Protein Assay Reagent Kit (Thermo Fisher Scientific, Inc., CA, USA). Samples containing 60 µg of total protein were separated by sodium dodecyl sulfate-polyacrylamide gel electrophoresis (SDS-PAGE) using 10% gels and transferred to polyvinylidene fluoride (PVDF) membranes (Hybond-P; GE Healthcare UK Ltd., Buckinghamshire, UK). Membranes were probed with glucose transporter type 4 (GLUT4) and β-actin antibodies. The signal intensities were quantitated using the NIH Image J software (http://rsb.info.nih.gov/nih-image).

Statistical analyses

Data are expressed as the means ± SE. Statistical analyses were performed using SPSS software. Analysis of variance (ANOVA) was performed for the comparisons between groups. Significant differences (P < 0.05) between means were determined using Dunnett’s multiple comparison tests.

Results

Nutritional components in steamed rice, yellow koji, white koji, and red koji

The nutritional components in the steamed rice, yellow koji, white koji, and red koji added to the HFD were investigated (Table 1). The carbohydrate content was lower in rice koji than in steamed rice, whereas the protein content was higher in rice koji. However, the total energy, fat, and ash were approximately the same among steamed rice, yellow koji, white koji, and red koji. All koji molds produced an abundance of glycolysis enzymes such as glucoamylase and α-amylase. Furthermore, white koji mold produced a large amount of citric acid during white koji manufacture. Thus, glucose and citric acid contents were measured. The glucose content in rice koji was 1,500–2,000-fold larger than that in steamed rice. The citric content of white koji was dramatically higher than in the other materials. Although yellow koji and red koji also contained much higher citric acid levels than did steamed rice, these values were one-tenth the amount of those found in white koji. Ingredient composition of the diets fed to mice is shown in Table 2.

Table 1 Nutrient components of steamed rice and rice koji.

Component	g/100 g fwa	g/100 g dwb	Energy (kcal/100 g fw)a	
	Moisture	Carbohydrate	Protein	Fat	Ash	Glucose	Citric acid		
Steamed rice	4.1	87.5	7.1	1.0	0.3	0.1	2.9	387	
Yellow koji	5.4	82.9	9.3	2.1	0.3	20.0	95.9	388	
White koji	8.5	80.5	9.2	1.5	0.3	18.8	3306.7	372	
Red koji	4.9	83.4	9.6	1.8	0.3	15.0	104.0	388	
Notes.

a fw, fresh weight.

b dw, dry weight.

Table 2 Nutrient components of the diets using in this study.

Dieta	g/100g fwb	Energy (kcal/100g fwb)	
	Carbohydrate	Protein	Fat		
HFD-rice	32.4	24.3	32	510	
HFD-Y	31.7	24.5	32	509	
HFD-W	32.0	24.5	32	510	
HFD-R	32.0	24.5	32	510	
Notes.

a HFD-rice. high fat diet (HFD) with steamed rice, HFD-Y; HFD with yellow koji, HFD-W; HFD with white koji, HFD-R; HFD with red koji.

b fw, fresh weight.

Food intake and body and tissue weights of mice

Food intake of CE-rice diet and HFD was measured and expressed gram per day. CE-rice diet is the powder form and tended to be spilled out as shown in the variability of the amount of food measured (Table 3). The food intake rates for the HFD groups were 2.4–3.1 g per day and the energy intakes did not differ significantly between the HFD groups (Table 3).

Table 3 Body weight, food intake, and tissue weight.

	HFD-rice	HFD-W	HFD-Y	HFD-R	CE-rice	
Initial body weight (g)	21.43 ± 0.35 a	21.35 ± 0.37 a	21.32 ± 0.35 a	21.49 ± 0.32 a	21.30 ± 0.33 a	
Food intake (g/day)	2.94 ± 0.14 a	2.43 ± 0.06 a	3.13 ± 0.18 a	2.77 ± 0.16 a	6.70 ± 2.29 b	
Body weight gain (g)	8.40 ± 0.43 a	6.60 ± 0.45 b	6.80 ± 0.36 b	6.74 ± 0.34 b	3.43 ± 0.28 c	
Feed efficiency	10.36 ± 0.58 a	9.64 ± 0.50 ab	8.10 ± 0.70 b	8.90 ± 0.53 ab	1.93 ± 0.18 c	
Liver weight (g)	1.14 ± 0.04 a	1.07 ± 0.02 a	1.13 ± 0.03 a	1.11 ± 0.03 a	1.14 ± 0.02 a	
Total white adipose tissue (g)	1.96 ± 0.10 a	1.53 ± 0.15 b	1.54 ± 0.12 b	1.38 ± 0.10 b	0.53 ± 0.04 c	
Visceral white adipose tissue (g)	0.60 ± 0.06 a	0.50 ± 0.07 a	0.50 ± 0.07 a	0.43 ± 0.05 a	0.19 ± 0.02 b	
Epididymal white adipose tissue (g)	1.36 ± 0.08 a	1.04 ± 0.09 b	1.04 ± 0.07 b	0.95 ± 0.07 b	0.34 ± 0.02 c	
Brown adipose tissue (g)	0.11 ± 0.01 a	0.08 ± 0.01 a	0.10 ± 0.01 a	0.10 ± 0.01 a	0.08 ± 0.02 a	
Notes.

Data are presented as means ± SE; n = 12. Values in a row with different superscripts differ significantly at P < 0.05 by Dunnett’s multiple comparison test (a–c). Feed efficiency was calculated as follow: (weight gain for 4 weeks/food intake for 4 weeks) × 100.

The body weight increased gradually over the course of the experiment for all diet groups (Table 3). In particular, compared with the CE-rice diet, the HFD caused a significant increase in body weight. In groups receiving HFD, all rice koji significantly suppressed weight gain compared to HFD-rice diet. However, feed efficiency (weight gain/food intake × 100) was significantly reduced in the yellow koji group compared to the HFD-rice diet group. Liver weights were similar among all groups (Table 3). It was implied that HFD diet for 4 weeks did not induce the fatty liver. The HFD significantly increased the weights of total white adipose tissue compared with the CE-rice diet (Table 3). Furthermore, compared to HFD-rice diet, all rice koji significantly suppressed the weight of total white adipose tissue and epididymal white adipose tissue, in a manner similar to weight gain.

We performed blood analysis and gene expression profiling for these groups because the feed type-related differences on the mice appeared as the body weight gain after 4 weeks.

Lipid contents, adiponectin, and leptin levels in the serum of mice

Serum TC, HDL, and TG levels were measured to investigate the effects of rice koji on lipid metabolism (Table 4). After 4 weeks, the HFD had increased TC and HDL levels compared with those in the CE-rice diet group, while the TG level remained unchanged. Red koji was previously reported to reduce the levels of plasma natural lipid and TC (Trimarco et al., 2011). Significant differences were not found in TC, HDL, and TG levels between the HFD diet groups in this experiment. However, red koji showed a tendency to reduce the TG level in serum, consistent with previous studies.

Table 4 Serum lipid, adiponectin, leptin, blood glucose, and insulin levels.

	HFD-rice	HFD-W	HFD-Y	HFD-R	CE-rice	
TC (mg/dl)	121.44 ± 6.60 a	125.84 ± 4.24 a	123.13 ± 3.96 a	122.83 ± 6.90 a	62.36 ± 2.77 b	
HDL (mg/dl)	83.99 ± 4.03 a	88.94 ± 4.85 a	82.10 ± 5.38 a	82.95 ± 3.48 a	49.02 ± 2.52 b	
TG (mg/dl)	81.74 ± 7.20 a	87.72 ± 8.99 a	87.16 ± 7.84 a	72.10 ± 12.70 a	97.91 ± 8.60 a	
Adiponectin (µ g/ml)	16.77 ± 1.68 a	16.04 ± 0.30 a	15.91 ± 0.77 a	15.28 ± 1.00 a	15.54 ± 1.17 a	
Leptin (ng/mL)	13.87 ± 1.87 a	8.19 ± 1.73 b	9.17 ± 2.46 ab	10.47 ± 1.65 ab	0.88 ± 0.13 c	
Blood glucose (mg/dl)	198.92 ± 5.24 a	178.50 ± 5.39 b	198.25 ± 5.03 a	176.67 ± 3.16 b	147.25 ± 7.25 c	
Insulin (ng/ml)	1.32 ± 0.10 a	1.02 ± 0.09 a	1.12 ± 0.16 a	0.92 ± 0.18 a	0.40 ± 0.07 b	
HOMA-R	0.59 ± 0.05 a	0.41 ± 0.05 b	0.55 ± 0.06 ab	0.43 ± 0.06 ab	0.13 ± 0.02 c	
Notes.

Values in a row with different superscripts differ significantly at P < 0.05 by Dunnett’s multiple comparison test (a–c). Data are presented as mean ± SE; n = 9–12.

Adiponectin decreases the plasma glucose level and enhances the utilization of fatty acids in muscle (Berg, Combs & Scherer, 2002; Nawrocki & Scherer, 2004). Leptin, which has an antifeeding effect, is secreted from adipocytes in response to the accumulation of lipids in cells (Campfield, Smith & Burn, 1996). Therefore, the influence of rice koji on the secretion of adiponectin and leptin was investigated (Table 4). Although red koji modestly decreased serum adiponectin, it did not significantly change in the experimental groups. The leptin level increased in the HFD groups; however, compared with the HFD-rice group, white koji significantly suppressed this increase.

Blood glucose and serum insulin levels in the mice

C57BL/6J mice receiving HFD have exhibited severe obesity, hyperglycemia, and hyperinsulinemia (Harte et al., 1999). Herein, we presumed that glucose metabolism was altered by obesity in mice that fed on an HFD for 4 weeks and we therefore measured the blood glucose and serum insulin levels in all mice. A significant increase in blood glucose levels was observed in HFD-fed mice compared to those fed the CE-rice diet after 4 weeks. However, the blood glucose level was significantly lower in the white and red koji groups than in the HFD-rice group (Table 4). The HFD containing rice koji contained a small amount of glucose, approximately 1.5∼2.0% (w/w). Thus, our results showed that white and red koji efficiently suppressed the high blood glucose level induced by HFD. Red-mold-fermented-products have been reported to attenuate the development of diabetes and alleviate hyperglycemia (Shi & Pan, 2010), and our experiments confirmed this effect of red koji. Although the serum insulin level was not different among the experimental groups, the white and red koji groups tended to have lower insulin levels than those of the other HFD groups. HOMA-R, which was used to assess insulin resistance, was significantly lower in the white koji group than in the HFD-rice group. Although the red koji group showed no significant differences in HOMA-R, it tended to have lower HOMA-R than those of the other HFD groups, being reduced to almost the same level as that of the white koji group.

Expression of the fatty acid metabolism enzymes in mice

Rice koji presumably affect fat metabolism. To determine the molecules involved in fatty acid metabolism in HFD mice upon rice koji feed, we measured the hepatic gene expression of several genes involved in fatty acid metabolism. Fatty acid synthase (FAS), acetyl-CoA carboxylase α (ACACα), acetyl-CoA carboxylase β (ACACβ), carnitine palmitoyltransferase 2 (Cpt2), acyl-Coenzyme A dehydrogenase, very long chain (Acadvl), acyl-Coenzyme A dehydrogenase, long-chain (Acadl), acyl-Coenzyme A dehydrogenase, medium chain (Acadm) and peroxisome proliferator-activated receptor α (PPARα) was investigated herein. In the liver, the expression of FAS, ACACα, Cpt2, Acadvl, Acadl, Acadm and PPARα genes was not significantly different after 4 weeks of HFD feeding (Fig. 1). While the expression level ACACβ in the CE-rice group was significantly higher than the other HFD groups, it was not significantly different between HFD groups.

Figure 1 Expression of fat metabolism-related genes in liver of mice fed experimental foods.

Gene expression level of fatty acid synthase (FAS), acetyl coenzyme A carboxylase α (ACACα), acetyl coenzyme A carboxylase β (ACACβ), carnitine palmitoyltransferase 2 (Cpt2), acyl-coenzyme A dehydrogenase, very long chain (Acadvl), acyl-coenzyme A dehydrogenase, long-chain (Acadl), acyl-coenzyme A dehydrogenase, medium chain (Acadm), peroxisome proliferator-activated receptor-α (PPARα) were analyzed by real-time PCR. Data are expressed as the mean ± SE (n = 9). Dunnett’s test was used for multiple comparisons between individual groups (∗P < 0.05). HFD-rice (black bar), HFD-W (white bar), HFD-Y (light gray bar), HFD-R (gray bar), and CE-rice (dark gray bar).

Effect of white and red koji on glucose uptake and GLUT4 protein expression in L6 myotube cells

It was reported that red mold-fermented products could delay the increase in the blood glucose levels in rats and that red koji protects against diabetes (Hsieh & Tai, 2003; Shi & Pan, 2010). This effect of red koji was also confirmed in this study and that of white koji was first recognized (Table 4). Skeletal muscle is the predominant tissue for insulin-stimulated glucose disposal (Koranyi et al., 1991). Glucose enters the muscle cell primarily by facilitated diffusion by utilizing glucose transporter carrier proteins. GLUT4 is the predominantly expressed glucose transporter isoform in muscle (Shepherd & Kahn, 1999), and whole body glucose disposal and GLUT4 expression are correlated in healthy subjects (Koranyi et al., 1991). Therefore, the glucose uptake activity and GLUT4 protein expression level were measured in muscle cells to investigate the mechanism by which white and red koji suppresses blood glucose. L6 myotube cells were exposed to various concentrations of white and red koji extracts or steamed rice extract. Treatment of L6 myotube cells with white and red koji resulted in a dose-dependent increase in 2-NBDG uptake, similar to that in insulin-stimulated cells (Fig. 2). We examined the expression of GLUT4 proteins in L6 myotube cells stimulated with white and red koji extracts. GLUT4 expression increased by approximately 1.5 and 1.6-fold in L6 myotube cells treated with the extract of white and red koji, respectively, compared with cells treated with steamed rice (Fig. 3).

Figure 2 Dose-dependent analysis of 2-NBDG uptake by white and red koji extracts.

L6 myotube cells were incubated with steamed rice, white koji and red koji extracts at each of the indicated dilutions for 5 min. Each extract was prepared as described in Methods and Materials. Control indicates the L6 myotube cell treated by deionized water instead of the extract. The extract was diluted in deionized water and the dilution ratio of the extract is indicated by the dilution factor (DF). Insulin (0.1 µM) was used in place of the extract as a positive control. The 2-NBDG compound was added for 30 min and uptake was measured. Data are expressed as means ± SE (n = 12). Different letters above the bars were significantly different at P < 0.05 by Dunnett’s multiple comparison (a–d).

Figure 3 Glucose transporter type 4 (GLUT4) protein levels in L6 myotube cells treated with each sample extracts.

(A) Western blot analyses of GLUT4 and β–actin in L6 myotube cells. L6 myotube cells were treated for 14 h with steamed rice, white koji, and red koji extracts diluted at a factor of 1,000. (B) The relative signal intensities of GLUT4 to β-actin protein were measured by NIH image J software. Data are expressed as means ± SE (n = 5). Letters above the bars indicate significant differences of P < 0.05 by Dunnett’s multiple comparison (a, b).

To reveal the key compounds in red and white koji, it was investigated the effect of monacolin K, monascin and citric acid on the induction of GLUT4 protein in L6 myotube cells. Monacolin K is one of the most important compounds in red koji. A recent study has shown that monascin, which is a yellow pigment produced by Monascus sp., prevents fatty acid accumulation in high-fat diet-fed mice (Hsu et al., 2014). Citric acid is one of most abundant compounds in white koji. The exposure concentration was determined the previous results and citric acid concentration in white koji. Because concentrated citric acid solution is strongly acidic, cell damage was expected because of the low pH. Therefore, in this experiment, we used citrate-Na buffer (pH 7.4) to investigate the effect of citrate molecule. Our results showed monacolin K and monascin can induce GLUT4 expression (Fig. 4). Citric acid did not have this effect on L6 myotube cells.

Figure 4 GLUT4 protein levels in L6 myotube cells treated with each compounds.

L6 myotube cells were treated with monacolin K, monascin, or citrate-Na buffer (pH 7.4). After 14 h, the expression of GLUT4 and β-actin proteins was measured by western blot analysis. The concentrations of each compound were shown as the final concentration in the medium. The relative signal intensities of GLUT4 to β-actin protein were measured by NIH image J software.

Discussion

We used an HFD-feeding animal model to evaluate and confirm the antiobesity effects of three types of rice koji: white koji, yellow koji, and red koji. All types of rice koji significantly decreased weight gain, epididymal white adipose tissue, and total adipose tissue without affecting food intake. It is known that obesity develops when energy intake exceeds energy expenditure. Feed efficiency was significantly reduced in the yellow koji group compared to the HFD-rice diet group. White and red koji showed also a tendency to reduce food efficiency. Taken together, our results suggest that white, yellow, and red koji have a protective effect on HFD-induced fat accumulation by increasing energy expenditure or inhibiting the absorption of excess fat in the diet. These results supported the hypothesis that the rice koji has the anti-obesity effect, and these effects of each rice koji appear to be triggered by different mechanisms because the different koji types have different propensity for feed efficiency, leptin level, blood glucose, and HOMA-R.

Red koji is reported to exhibit a protective effect on alcoholic liver disease, including fatty liver (Cheng & Pan, 2011). We have demonstrated herein that rice koji containing red koji protect against diet-induced obesity. Furthermore, white and red koji was effective in reducing blood glucose level compared to the HFD-rice diet. It was confirmed that white and red koji increased 2-NBDG uptake and GLUT4 protein expression in L6 myotube cells. It was also revealed that monacolin K and monascin induced GLUT4 expression on L6 myotube cells. Therefore these compounds could contribute to the increase of GLUT4 protein and the acceleration of glucose uptake by red koji. Although we failed to show the difference between the experimental groups of GLUT4 expression in gastrocemius muscle tissue of mice (data not shown), it might be difficult to detect the difference without separation of red from white muscle tissue, the former of which contains more GLUT4 protein (Kern et al., 1990).

White koji contains a high amount of citric acid in contrast to yellow and red koji. Citric acid, a natural and dietary chelator found in citrus fruits, is widely used in food products as a preservative. Furthermore, it plays a key role in the TCA cycle, which is part of the metabolic pathway involved in the chemical conversion of carbohydrates, fats, and proteins. It was recently reported that citric acid prevents cataracts and nephropathy in diabetic rats (Nagai et al., 2010); this suggests that the energy flux was improved by the citric acid in white koji. Therefore, we expected that citric acid was one of the key compounds in white koji and examined the effect on the induction of GLUT4 protein in the muscle cells. However it did not have the ability. We will continue to look for the key compounds in white koji except for citric acid, because Aspergillus kawachii which is the koji mold used for white koji preparation do not produce monacolin K and monascin.

Sake is made from fermented rice, yellow koji, and yeast. Sake cake, which is a residue of sake mash, has also been reported to be effective in lowering cholesterol levels in rats (Ashida et al., 1997). The solid content in sake cake has over 70% protein content, and more than 90% of its protein is derived from rice. As sake cake is similar to casein I in its digestive efficiency, the indigestable rice protein changes its form into one that can be easily digested by digestive enzymes. Furthermore, pepsin-digestive sake cake is more effective in increasing fecal lipid levels than sake cake and tends to reduce the serum and liver cholesterol and lipid levels (Ashida et al., 1997). Thus, digested rice protein is more effective than its indigestable form in reducing the intestinal lipid absorption and liver and serum cholesterol levels. All rice koji types contain proteolytic enzymes. Digested rice protein is considered to be one of the common functional compounds in rice koji. Furthermore, it was previously reported that the amino acid composition of plant protein in diets also affects the cholesterol level (Vahouny et al., 1985), and rice koji contains compounds from fungus. These compounds may function together in a coordinated manner although further studies are necessary to elucidate the exact mechanisms.

Rice koji accounts for 7% of the total calories of the diets used in this study. It may be difficult for human beings to consume a comparable level of rice koji. However, in Japan, rice koji could be consumed in dairy meals in various forms. For example, rice koji-miso is prepared with soy and rice koji in a 1:1 proportion, and people can consume this preparation, such as through miso soup. Amazake is saccharified rice koji prepared by incubating rice koji overnight with water at approximately 55 °C; it is also a traditional Japanese drink. Because of this, rice koji is considered a readily accessible functional food and is gaining attention as a natural source of nutrients and health-promoting compounds.

In conclusion, we showed all rice koji had the anti-obesity effect by the mechanisms other than food intake regulation. It may involve feed efficiency but may differ depending on the type of rice koji. White and red koji may improve glucose tolerance by the activation of glucose uptake through the increase of GLUT4 protein expression in the muscle. Rice koji should be considered a readily accessible functional food that may be effective for the prevention or treatment of metabolic syndrome.

Supplemental Information

Supplemental Information 1 Raw data

Raw data for all figures and tables.

Click here for additional data file.

The authors thank K Kuwamoto MSc. (Faculty of Agriculture, Kagoshima University) for providing technical assistance.

Additional Information and Declarations

Competing Interests

Author Contributions

Animal Ethics

Akio Inui is an Academic Editor for PeerJ.

Yumiko Yoshizaki performed the experiments, analyzed the data, contributed reagents/materials/analysis tools, wrote the paper, prepared figures and/or tables.

Chihiro Kawasaki, Kai-Chun Cheng, Miharu Ushikai and Kayu Okutsu performed the experiments, analyzed the data, contributed reagents/materials/analysis tools.

Haruka Amitani wrote the paper.

Akihiro Asakawa conceived and designed the experiments, performed the experiments, analyzed the data, contributed reagents/materials/analysis tools.

Yoshihiro Sameshima conceived and designed the experiments.

Kazunori Takamine and Akio Inui reviewed drafts of the paper.

The following information was supplied relating to ethical approvals (i.e., approving body and any reference numbers):

This study was approved by the ethical committee for animal experimentation at Kagoshima University Graduate School of Medical and Dental Sciences (Permit Number: MD10122).

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
