# Peer review of "Rice koji reduced body weight gain, fat accumulation, and blood glucose level in high-fat diet-induced obese mice"

_PeerJ, doi:10.7717/peerj.540_

## Round 0.1 · original submission · Minor Revisions

Reviewer 1 recommends to accept your manuscript in the current form for publication in PeerJ, but Reviewer 2 recommends some revision before acceptance for publication of your manuscript in PeerJ.

I hope that you could revise your manuscript according to the Reviewer's comments by providing additional experimental results or some discussion resolving points raised by Reviewer 2.

Reviewer 1 ·

Basic reporting

Very good.

Experimental design

Excellence.

Validity of the findings

No problem.

Additional comments

I think that it is a excellent article.

Reviewer 2 ·

Basic reporting

The authors demonstrated that all rice koji have the anti-obesity or anti-diabetes effects although the mechanisms may differ depending on the type of rice koji consumed. This study include an interesting finding, hover, they should revise it according to the following suggestions.

Experimental design

1) To clarify the effects of these extract on hepatic fatty acid synthesis, the authors should measure the expression of mRNA levels of ACD, acyl-coenzyme A dehydrogenase, CPT II, carnitine palmitoyl transferase II, ACC, acetyl-CoA carboxylase, ACL, ATP citrate lyase in addition to FAS, ACACbeta, and PPARalpha.
2) The authors should investigate the GLUT4 expression on muscle of in vivo model, and should investigate the effect of these extracts on the expression of GLUT4.

Validity of the findings

The authors should carefully evaluate the effects of extracts on myotube cells, and also should confirm these findings using samples from in vivo model.

Additional comments

They should revise it according to the following suggestions.

---

## Round 0.2 · accepted · Accept

I read your revised manuscript. I recognize that you have appropriately revised the manuscript according to the reviewers' comments. Therefore, I have no more comments on your revised manuscript, which is accepted for publication in PeerJ.
Thank you very much for your submission to PeerJ.